# Dental caries and associated factors among patients visiting Shashamane Comprehensive Specialized Hospital

**Aliye Geleto[1], Edao Sinba[2], Musa Mohammed Ali[3]\***

1 Dental Clinic, Shashamane Comprehensive Specialized Hospital, Shashamane, Ethiopia, 2 Department of Public Health, Shashamane Campus, College of Health Science, Madda Walabu University, Shashamane, Ethiopia, 3 School of Medical Laboratory Science, College of Medicine and Health Sciences, Hawassa University, Hawassa, Ethiopia

\* ysnmss@yahoo.com

## Abstract

### Introduction

Dental caries is a major public health problem. In Ethiopia, prevention and treatment of oral health-related illness had given little attention and there is limited data on the extent and factors associated with oral health-related illnesses and oral care practices.

### Objective

This study was aimed to investigate the prevalence of dental caries and associated factors among patients visiting Shashamane Comprehensive Specialized Hospital (SCSH).

### Methods

A hospital-based cross-sectional study was conducted among 288 patients who visited SCSH dental clinic from March 1, 2021, to April 15, 2021. A questionnaire was employed to collect the background characteristics of the participants. Dental caries was confirmed as per World Health Organization guidelines. Data were analyzed using SPSS version 24. Bivariable and multivariable logistic regression were used to determine predictors of dental caries. A $p$-value less than 0.05 was taken as a cut point to determine a significant association.

### Results

The overall prevalence of dental caries was 64.6% with 95% CI (58.8–70.1). The mean of Decayed, Missing, and Filled Teeth was 1.33. Dental caries was significantly higher among respondents who did not brush their teeth (AOR = 3.589, 95% CI:1.756–7.334), who consumed sugary food (AOR = 3.650, 95% CI: 1.747–7.628), those with monthly a income of less than 5000.00 Ethiopian Birr (AOR = 2.452, 95% CI (1.193–5.042), and those who had poor oral hygiene status (AOR = 1.826, 95% CI: 0.901–3.700).

**Data Availability Statement:** All relevant data are within the manuscript.

**Funding:** The authors received no specific funding for this work.

**Competing interests:** The authors have declared that no competing interests exist.

**Abbreviations:** DMFT, Decayed, Missed and Filled Teeth; SCSH, Shashamane Comprehensive Specialized Hospital; WHO, World Health Organization.

## Conclusions

This study revealed a high prevalence of dental caries among patients visiting the dental clinic. Tooth brushing habits, consumption of sugary food, and poor oral hygiene were significantly associated with dental caries.

## Introduction

The disease of the oral cavity including dental caries is a huge health problem throughout the globe and people of any age can be affected. The oral cavity-related disease can cause pain, discomfort, disfigurement, and sometimes death. Approximately, 3.5 billion people are affected by oral cavity-related diseases [1, 2].

Dental caries develops over time and it has multiple causative agents. Dietary habits, which influence normal flora of the oral cavity, along with host susceptibility is the main factor for dental caries development. Although not often life-threatening, dental caries represents a major public health problem because of its high prevalence and significant impact on general health [3]. Based on the Global Burden of Disease, dental caries (untreated) of permanent teeth is the most predominant condition. About half a billion children are affected by dental caries of milk teeth [4].

There is a piece of evidence to prove the interrelationship between oral and general health [5]. Most systemic diseases such as diabetes and the heart-related problem may have oral signs and symptoms. Even though there is improvement in maintaining oral health worldwide, the problem is still predominant in list-income countries [6]. Dental caries has historically been considered an important component of the global disease burden, which can be effectively prevented and controlled through a combination of community, professional, and individual actions. Early detection of disease in most cases is crucial to control oral-related health problems. Worldwide, the prevalence of dental caries among adults is high as the disease affects nearly 100% of the population in most countries [6].

In Africa, less attention is given to oral health as a result there is limited data in this regard. However, non-communicable diseases, including oral conditions are becoming more prevalent; and there are substantial oral health inequalities in both high and low-income nations across the continent including Ethiopia [7].

Management of oral-cavity-related disease including dental caries is expensive and most of the time it is not considered as part of general health. The cost of dental treatment in most developed countries accounts for about 5% of total health costs [4]. Developing countries are unable to provide services related to oral health diseases. Several factors contribute to oral diseases development which includes consumption of sugary food, smoking cigarette, and consumption of alcohol. The majority of oral diseases can be treated and prevented at early stages [4].

Considering the paucity of literature on dental caries and the public health importance of dental caries, this study is planned to provide some information on the prevalence of dental caries and associated factors among patients attending Shashamane Comprehensive Specialized Hospital (SCSH) dental clinic.

## Methods

### Study area and design

A hospital-based cross-sectional study was conducted from March 1, 2021, to April 15, 2021, at SCSH Dental Clinic. SCSH provides health care services for over 2 million people. The

hospital is located 245 km to the south of Addis Ababa, the capital of Ethiopia, in the West Arsi zone, Oromia regional state. The dental clinic is one of the specialty clinics in the hospital that provides inpatient and outpatient dental services. The average patient flow of the dental clinic per month was 600.

## Source population

All patients who attended SCSH dental clinic during the study period were considered as source population whereas all systematically selected patients who visited SCSH dental clinic during the study period were taken as the study population.

## Eligibility criteria

All patients who visited the dental clinic and aged greater than one year were included. Patients who were critically sick and unable to communicate were excluded from the study.

## Study variables

**Dependent variables.** Dental caries.

**Independent variables.** Socio-demographic characteristics: Age, Sex, Educational status, Marital status, Monthly income, Residency.

Dietary habits: Use of sugary foods, Smoking cigarettes, Chewing chat.

Oral hygiene: Tooth brushing habit, Tooth brushing method, Tooth brushing frequency, Fluoride toothpaste usage.

## Sample size determination

The sample size was calculated using single population proportion formula with the following assumption, 5% margin of error, 95% Confidence level, 78.2% prevalence of dental caries obtained from a similar study conducted at Debre Tabor General Hospital [8]. After considering 10% for non-response rate the total sample size was 288. Study participants were selected by systematic random sampling technique; with a sampling interval of two until the desired sample size of 288 was reached (Interval size, K = N/n = 600/288 ~ 2).

## Data collection

A structured questionnaire modified from the WHO oral health survey [9] was used to collect socio-demographic characteristics, dietary habits, healthcare-seeking behavior towards oral health problems, and factors that affect dental health. Socio-demographic characteristics include age, sex, monthly income, place of residence, educational status, and marital status. Dietary habits and healthcare-seeking behavior include sugary food consumption and dentist visits respectively. Factors that affect dental health include smoking cigarettes, tooth brushing habits, oral hygiene, and chewing of khat.

## Physical examination

Each tooth of the individuals was examined for dental caries according to WHO recommendation [10]. Dental probes, dental mirrors, disposable tongue depressors, and flashlights were used to perform the physical examination. Teeth were inspected and probed for the presence of Decay (D), Missing (M), and Filled (F). Clinically, caries was confirmed if a lesion is seen in a pit or fissure, or on the tooth surface, and presence of detectable softened floor, softened wall, or undermined enamel. A filled tooth that contains one or more restorations and one or more areas that are decayed was also included in this category. If any doubt existed, caries was

not recorded as present. The tooth was considered as missing because of caries if a person had pain history and/or with cavity before extraction. All clinical observations were recorded on the assessment form.

## Data quality control

A questionnaire was originally developed in English and translated to Amharic and Afaan Oromo and translated back to English to confirm its consistency. The pretest was done at Melka oda general hospital, Shashamane, Ethiopia among individuals representing 5% of the total sample size ahead of the study. The aim of the pretest was to check the suitability and consistency of the data collection tool. Training on how to conduct the interview was given for one day for the interviewers. There was regular supervision during data collection. At the end of each working day, completed questionnaires and physical examination forms were checked for completeness and accuracy of the records. The questionnaire was checked for clarity, sensitivity to culture, and the presence of appropriate words to the community. Multicollinearity diagnosis was performed using all indicators to assess the interaction between independent variables.

## Data analysis

The completed questionnaire was checked for completeness and consistency of the responses. Then, data were entered to EPI INFO 7and cleaned then analyzed using statistical package for social science (SPSS) version 24. Frequency and percentage were computed using univariate analysis to get summary values. Binary logistic regression was done and variables with a $p$-value < 0.25 were selected for multivariable analysis. The possibility of multicollinearity was checked before running multivariable logistic regression. Considering all indicators used to diagnose multicollinearity together; variables that had variance inflation factor (VIF) greater than 10 (1/ $(1-R^2)$, tolerance less than $0.1(1-R^2)$, condition index greater than 50 (or 30), Eigen value less than 0.01, and Proportion of Variation greater than 0.8 (or 0.7) were excluded from multivariate analysis. Factors with a $p$-value less than 0.25 were analyzed using Multivariable logistic regression to determine predictors of dental caries. Significance was set at $p<0.05$.

## Operational definitions

**Dental caries.** The presence of tooth decay, missing and filled teeth at the time of oral examination.

**Good oral hygiene.** If no food particles and no accumulation of dental plaque or calculus are visible on the tooth surface at the time of oral examination.

**Poor oral hygiene.** If there are food particles in the mouth and there is visible plaque/calculus at the time of oral examination.

**Decayed tooth (D).** Includes carious teeth, filled teeth with recurrent decay, a tooth with the only root left, temporarily filled teeth surface with other surfaces (parts) decayed.

**Missed tooth (M).** Includes tooth that is missed due to caries but it doesn't include teeth missing for reasons other than caries.

**Filled teeth (F).** Includes teeth that have one or more permanent restoration with no secondary (recurrent) caries [8].

**The DMFT index:** is the average number of teeth per person that are decayed (D), missed because of caries (M), or filled (F). It is a quantitative expression of the lifetime caries experience of the permanent teeth. In the calculation of the DMFT index, the numerator is the sum total of DMF teeth and the denominator is the total number of persons examined.

## Ethical approval and consent to participate

Before initiation of the current study, ethical clearance and permission were obtained from the Institutional Review Board (IRB) of Oromia Health Bureau (Reference number: BEFO/HBTFU/1-16/1078) and SCSH. Written informed consent and assent for participants aged below 18 was obtained before data collection. Written informed consent was also obtained from parents or guardians of children. Any information which indicates the identity of the participants was removed from the data collection tool. All information collected from participants was kept confidential. All procedures were performed in accordance with the Declaration of Helsinki.

## Results

### Socio-demographic characteristics

From the total of 288 participants, all of them responded making a response rate of 100%. 136 (47.2%) participants were females and nearly two-thirds of the respondents 196 (68.1%) attended formal education. The mean age of the respondent was 31.7 with ±13.4 standard deviations (SD) and 107 (37.2%) of the respondents were in the age group of 20 to 29 years. 197 (68.4%) of the respondents were married. 172 (59.7%) of the respondents had a monthly income of less than 5000.00 Ethiopian birrs (Table 1).

### Prevalence of dental caries

The overall prevalence of dental caries found in this study was 64.6% with 95% CI (58.8–70.1). The mean Decayed, Missing, and Filled Teeth (DMFT) were 1.33.

### Factors associated with dental caries

The present study revealed a significant association between respondents' level of education and dental caries (AOR = 0.025, 95% CI: 0.006–0.095). Dental caries was 3.5 times higher among respondents who did not brush their teeth as compared to those who brushed their teeth (AOR = 3.589, 95% CI: 1.756–7.334). A participant who consumed sugary foods had a

**Table 1. Socio-demographic characteristics of patients attending the dental clinic in Shashamane Comprehensive Specialized Hospital, southeast Oromia, Ethiopian, 2021 (N = 288).**

| Category | | Frequency | Percent |
|---|---|---|---|
| Age group in years | <20 | 39 | 13.5 |
| | 20–29 | 107 | 37.2 |
| | 30–39 | 74 | 25.7 |
| | 40–49 | 32 | 11.1 |
| | 50–59 | 21 | 7.3 |
| | ≥60 | 15 | 5 |
| Sex | Male | 152 | 52.8 |
| | Female | 136 | 47.2 |
| Educational status | Attended formal education | 196 | 68.1 |
| | Not attended formal education | 92 | 31.9 |
| Residence | Urban | 103 | 35.8 |
| | Rural | 185 | 64.2 |
| Marital status | Married | 197 | 68.4 |
| | Single | 78 | 27.1 |
| | Divorced | 13 | 4.5 |

3.6 times higher chance of having dental caries than those who did not consume sugary foods (AOR = 3.650, 95% CI: 1.747–7.628). Respondents who had poor oral hygiene status were 1.8 times more likely to be affected by dental caries as compared to those patients whose oral hygiene status was good (AOR = 1.826, 95% CI: 0.901–3.700). Dental caries was lower among respondents who did not chew khat than those who chew khat (AOR = 0.279, 95% CI: 0.127–0.611) and dental caries was 2.4 times higher among respondents who earned less than 5000.00 ETB per month as compared to those who earned greater 5000.00 ETB (AOR = 2.452, 95% CI: 1.193–5.042) (Table 2).

## Discussion

There is limited data on dental caries, especially in Southern parts of Ethiopia. We attempted to assess the prevalence and associated factors of dental caries among patients attending the SCSH dental clinic. This study revealed a high prevalence of dental caries, 64.6%, which is comparable with a cross-sectional study conducted in Turkey (62%) that involved large sample size [11]; however, it is higher than school-based cross-sectional studies conducted in Kenya 37.5% [12], Bahirdar, Ethiopia (21.8%) [13], North West, Ethiopia (36%) [14], and a hospital-based cross-sectional-study conducted in Gondar, Ethiopia (23.64%) [15]. A systematic review and meta-analysis conducted in East Africa (45.7%) [16] and Ethiopia (40.98%) [17] and a school-based cross-sectional study conducted in Shawa, Ethiopia [18] also revealed a low prevalence of dental caries as compared to the current study.

The lowest prevalence of dental caries was reported from Tanzania (8.8%) which is a school-based cross-sectional study) [19], and a high prevalence was reported from Sudan (87.7%) [20]. However, our finding is lower than community-based cross-sectional-study

**Table 2. Bivariate and multivariable analysis of factors that can predict dental caries among patients attending the dental clinic of Shashamane Comprehensive Specialized Hospital.**

| Variables | Category | Dental caries (n, %) | | COR (95% CI) | p-Value | AOR (95% CI) | p-value |
|---|---|---|---|---|---|---|---|
| | | Yes | No | | | | |
| Educational status | Not attended | 89 (96.7) | 3(3.3) | 0.033(0.010–108 | 0.000 | 0.025 (0.006–0.095) | <0.001* |
| | Attended | 97 (49.5) | 99(50.5) | 1 | | 1 | |
| Khat chewing | Yes | 80 (79.2) | 21(20.8) | 0.344(0.196–602) | 0.000 | 0.279 (0.127–0.611) | 0.001* |
| | No | 106(56.7) | 81(43.3) | 1 | | 1 | |
| Sugary food consumption | Yes | 131(76.6) | 40(23.4) | 3.692(2.223–.130) | 0.000 | 3.650 (1.747–7.628) | 0.001* |
| | No | 55(47.0) | 62(53.0) | 1 | | 1 | |
| Tooth brushing habit | Yes | 56(43.1) | 74(56.9) | 1 | | 1 | |
| | No | 130(82.3) | 28(17.7) | 0.163(0.095–0.277) | 0.000 | 3.589(1.756–7.334) | <0.000* |
| Oral hygiene | Good | 40(41.7) | 56(58.3) | 3.625 (2.185–6.016) | 0.000 | 1.826 (0.901–3.700) | 0.005* |
| | Poor | 146(76.0) | 46(24.0) | 1 | | 1 | |
| Monthly Income | <5000.00 ETB | 112(67.5) | 54(32.5) | 1.862 (1.139–3.043) | 0.013 | 2.452 (1.193–5.042) | 0.015* |
| | ≥5000.00 ETB | 74(60.7) | 48(39.3) | 1 | | 1 | |
| Dentist visit | Yes | 117(70.9) | 48(29.1) | 1.908(1.169–3.112) | 0.010 | 2.106 (0.891–4.976) | 0.090 |
| | No | 69(56.1) | 54(43.9) | | | 1 | |
| Smoking cigarette | Yes | 72(74.2) | 25(25.8) | 1.945 (1.135–3.335) | 0.016 | 3.773(0.767–18.563) | 0.102 |
| | No | 114(59.7) | 77(40.3) | 1 | | 1 | |
| Residence | Rural | 76(75.2) | 25(24.8) | 0.417(0.242–0.717) | 0.002 | 0.867 (0.374–2.012) | 0.740 |
| | Urban | 110(58.8) | 77(41.2) | 1 | | 1 | |

*Statistically significant association, ETB: Ethiopian Birr

done in Indonesia (82.4%) [21] and a cross-sectional study conducted in Debre Tabor, Ethiopia (78.2%) [8]. The variation observed could be due to the nature of the population studied. For instance, a study conducted in Kenya considered 12 years old children which attended public primary schools [12] whereas a study conducted in Turkey included adults aged 18–74 years that were invited to a dental clinic for free dental examination [11]. A study from Sudan was population-based considered individuals aged ≥16 [20]. As this is a hospital-based study, there might be a high patient flow that could have resulted in a high prevalence of dental caries. The difference with the Kenya and Indonesia studies might be due to heterogeneity of study population and, the socio-demographic differences between those countries.

The MDFT index detected in this study, 1.33, was comparable with a study conducted in North Showa, Ethiopia among school children (1.28±1.21) [18] and a study from Gondar Ethiopia (1.095±0.24) [15]. A systematic review and meta-analysis conducted by Teshome et al. [16] reported a comparable mean of DMFT (1.9441). However, Teshome et al. [16] reported a high mean of DMFT from Sudan (3.146) and Uganda (2.876). The highest mean of DMFT was reported from Sudan (8.7+5.9) [20] and the lowest mean of DMFT was reported from Kenya (0.24) [22]. Countries that reported a higher mean of DMFT might be due to poor oral hygiene practice and regular consumption of sugary foods.

In this study, factors such as educational status, tooth brushing habit, sugary food consumption, khat chewing habit, monthly income, and oral hygiene status showed significant association with the prevalence of dental caries. The higher prevalence of dental caries in those who did not attend formal education than their counterparts could possibly be due to the indirect effect of education on dental caries. Those who attended formal education might have awareness of dental caries and take regular appropriate measures to prevent dental caries. An educated person can read and obtain information about oral health while those who are not educated may not know the cause of dental caries and the measures to be taken for its prevention. Moreover, information related to oral health might be given during formal education. This finding was also supported by similar studies conducted in Gondar, Ethiopia [14, 15] and Turkey [11]. This study revealed a strong positive association between caries development and tooth brushing practice. Participants who did not brush their teeth were 3.5 times at a higher risk of dental caries. A similar finding was reported from Bahirdar, Ethiopia [13] and Indonesia [21]. This could be due to the washing away of sugary food ruminants from the teeth during tooth brushing; therefore, micro-organisms will not get enough time to establish themselves and produce acid which is responsible for caries development [23].

This study showed that patients who did not chew khat had less chance of developing dental caries as compared to those who consumed khat. This could be elaborated as those who chew khat has a tendency to use soft drinks and sugars that help the growth of bacteria on the tooth surfaces. Khat is commonly used in Ethiopia for social and pleasure purposes. Also, certain occupational groups like long-distance truck drivers and students during examination time have a high tendency to consume Khat.

Respondents who had poor oral hygiene status were about 1.8 times more likely to develop dental caries. This finding was in line with a study conducted in south Gondar [8, 15]. Poor oral hygiene increases *Streptococcus mutans* colonization and in severe cases, the loss of enamel [13]. Poor oral hygiene can lead to the build-up of harmful plaque-forming bacteria that demineralizes teeth enamel and cause dental caries [3]. Respondents who consume sugary foods were 3.65 times at higher risk of dental caries development as compared to those who did not consume sugary foods. This finding was in agreement with the study conducted in Kenya [12, 17]. This might be due to sugary food that facilitates the growth of *Streptococcus mutans* which can produce acid and lead to the development of dental caries and ultimate tooth decay [24].

Even though statistically not significant, a large proportion of patients who smoke cigarettes had dental caries when compared to those who didn't smoke cigarettes. This is supported by one study conducted in Portugal, that confirms smoking cigarettes as a risk factor for dental caries [19]. Smoking cigarettes can facilitate bacterial growth and acid production by reducing the buffering capacity of saliva [20].

There was also a significant association between the monthly income of participants and dental caries. Dental caries was 2.452 times higher among respondents who earned less than 5000.00 ETB as compared to those who earned greater than 5000.00 ETB. This finding is in line with the study done in Gondar, Ethiopia [14]. As the income increase, the chance of developing dental caries will decrease. This might be due to study participants with high monthly income are able to buy tooth cleaning materials and visit the dental clinic regularly. The burden of dental caries can be reduced by providing proper health education on how to keep oral hygiene and regular visit to dental clinic.

## Limitation of the study

To detect dental caries, we used clinical diagnosis only; it was not supplemented with microbiological methods. The difficulty of radiological examination due to the lack of instruments and laboratory setup might reduce the actual magnitude of the problem. Sweet food items and drinks were assessed by the usual patterns of intake but the amount and the duration of intake were not assessed. The monthly income of participants was assessed by usual estimation as the majority of respondents were from rural areas that have no constant monthly income and this might also hide the true monthly income. The determinant factors were not exhaustive. There could be other determinant factors of dental caries that our study did not address.

## Conclusions

The prevalence of dental caries in the current study was high. Educational status, tooth brushing habits, consumption of sugary food, khat chewing, monthly income, and oral hygiene status were significantly associated with the prevalence of dental caries. It is important to provide health education on how to improve horal health in the study area.

## Acknowledgments

We would like to thank all staff members of Shashamane Comprehensive Specialized Hospital dental clinic for their assistance during data collection. We acknowledge all study participants for their willingness to take part in the study.

## Author Contributions

**Conceptualization:** Aliye Geleto, Edao Sinba, Musa Mohammed Ali.

**Data curation:** Aliye Geleto, Edao Sinba.

**Formal analysis:** Aliye Geleto, Edao Sinba.

**Investigation:** Aliye Geleto.

**Methodology:** Aliye Geleto, Edao Sinba, Musa Mohammed Ali.

**Supervision:** Edao Sinba.

**Validation:** Edao Sinba.

**Visualization:** Edao Sinba, Musa Mohammed Ali.

**Writing – original draft:** Musa Mohammed Ali.

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
