## [Decision Letter · Decision Letter 0]

2 Feb 2022

PONE-D-21-30095Dental caries and associated factors among patients visiting Shashamane Comprehensive Specialized HospitalPLOS ONE

Dear Dr. Ali,

Thank you for submitting your manuscript to PLOS ONE. After careful consideration, we feel that it has merit but does not fully meet PLOS ONE’s publication criteria as it currently stands. Therefore, we invite you to submit a revised version of the manuscript that addresses the points raised during the review process.

We look forward to receiving your revised manuscript.

Kind regards,

Frédéric Denis, Ph.D.

Academic Editor

PLOS ONE

Journal Requirements:

3. Please include additional information regarding the survey or questionnaire used in the study and ensure that you have provided sufficient details that others could replicate the analyses. For instance, if you developed a questionnaire as part of this study and it is not under a copyright more restrictive than CC-BY, please include a copy, in both the original language and English, as Supporting Information

not applicable

NO authors have competing interests

Reviewers' comments:

Reviewer's Responses to Questions

**Comments to the Author**

1. Is the manuscript technically sound, and do the data support the conclusions?

Reviewer #1: Yes

Reviewer #2: Partly

2. Has the statistical analysis been performed appropriately and rigorously? 

Reviewer #1: I Don't Know

Reviewer #2: N/A

3. Have the authors made all data underlying the findings in their manuscript fully available?

Reviewer #1: No

Reviewer #2: Yes

4. Is the manuscript presented in an intelligible fashion and written in standard English?

Reviewer #1: Yes

Reviewer #2: No

5. Review Comments to the Author

Reviewer #1: Abstract : within 300 words

Line no 16-17: In Ethiopia, prevention and treatment of oral health-related illness had been given little attention line no 33: This study revealed a high prevalence of dental carries

Line no 36-37: Key words should be arranged alphabetically

Line no 52 and 53: Even though there is improvement in maintaining oral health worldwide, the problem is still predominant in low-income countries. Also reference for this line.

Line no 204-6 : rephrasing required.

Recommended read:

1. Zewdu T, Abu D, Agajie M, Sahilu T. Dental caries and associated factors in Ethiopia: systematic review and meta-analysis. Environ Health Prev Med. 2021 Feb 12;26(1):21. doi: 10.1186/s12199-021-00943-3. PMID: 33579186; PMCID: PMC7881546.

2. Teshome A, Andualem G, Derese K. Dental Caries and Associated Factors Among Patients Attending the University of Gondar Comprehensive Hospital Dental Clinic, North West Ethiopia: A Hospital-Based Cross-Sectional Study. Clin Cosmet Investig Dent. 2020;12:191-198

https://doi.org/10.2147/CCIDE.S247179

3. Anguach Shitie, Rahel Addis, Abebe Tilahun, Wassie Negash, "Prevalence of Dental Caries and Its Associated Factors among Primary School Children in Ethiopia", International Journal of Dentistry, vol. 2021, Article ID 6637196, 7 pages, 2021. https://doi.org/10.1155/2021/6637196

Reviewer #2: Thank you for the manuscript.

First of all, the anonymity of survey data is not mentioned. Also, what steps did the authors took to safeguard the patient identity.

In 104 line it was mentioned that a structured questionnaire was modified from WHO oral health survey. Could you please elaborate on that? Furthermore, the exact questionnaire should be presented in the manuscript.

How many evaluators were there? Were they calibrated?

The uniqueness of the population studied with respect to the other African studies was not discussed in detail. The introduction does not describe in detail the main issue of the paper.

lines 234-239 - there is no literature reference

line 116 - if the tooth was extracted not at the same clinic and the authors did not have any records of that, how did authors determine the reason of tooth extraction?

The authors said that the response rate was 100% and it is hard to believe that all of 288 patients have fully filled the questionnaires.

lines 166 - 167 - the authors should change the words into numbers, there has to be consistency throughout the manuscript.

There cannot be a p value equal to 0.000.

What do the authors consider as a statistically significant result? When COR is <0.05, AOR <0.05 or both?

The discussion section is descriptive, and does not really discuss the results. The results were compared to other studies, however, there were no suggestions how to solve the problems. In the conclusions there are new ideas like integrating promotion of oral health services that has been never mentioned in the discussion, therefore, the conclusions does not really conclude the manuscript. Furthermore, when the results are compared to other studies - where the studies conducted in other countries in the same design? A specification is needed.

The quality of the language could be better. The authors change past and present times a few times throughout the manuscript.

line 43 - develops instead of develop

56 - ‘to’ is missing

69 - keeping in mind - lowercase letter should be changed in to uppercase at the beginning of the sentence

179- index is not a DMF, but DMFT, I would recommend using the newest D3MFT index

183- didn’t into did not

211- MDFT index

Throughout the manuscript different forms of words used - ‘physical examination’ vs ‘intraoral examination’ and they should be unified.

6. PLOS authors have the option to publish the peer review history of their article (what does this mean?). If published, this will include your full peer review and any attached files.

Reviewer #1: No

Reviewer #2: No

---

## [Author Response · Author response to Decision Letter 0]

7 Feb 2022

Response to comments

Journal Requirements:

Response: Thank you for the comment; we have checked and corrected according it to the guideline. 

Response: We have moved Ethical consideration to Method section. We have also included a statement which indicates about parent or guardian consent for children.

3. Please include additional information regarding the survey or questionnaire used in the study and ensure that you have provided sufficient details that others could replicate the analyses. For instance, if you developed a questionnaire as part of this study and it is not under a copyright more restrictive than CC-BY, please include a copy, in both the original language and English, as Supporting Information

Response: The questionnaire (data collection tool) was adapted with modification from the WHO survey tool as indicated in section ‘data collection’ . We have included the reference for it. 

not applicable

NO authors have competing interests

Response: We moved the ethical consideration to the method section as mentioned above. 

Reviewers' comments:

Reviewer's Responses to Questions

Comments to the Author

1. Is the manuscript technically sound, and do the data support the conclusions?

Reviewer #1: Yes

Reviewer #2: Partly

2. Has the statistical analysis been performed appropriately and rigorously?

Reviewer #1: I Don't Know

Reviewer #2: N/A

3. Have the authors made all data underlying the findings in their manuscript fully available?

Reviewer #1: No

Reviewer #2: Yes

Response: We included all the data collected in the result section in the form of tables and text. 

4. Is the manuscript presented in an intelligible fashion and written in standard English?

Reviewer #1: Yes

Reviewer #2: No

Response: We have revised the English as shown in the manuscript with track change. 

5. Review Comments to the Author

Reviewer #1: Abstract : within 300 words

Response: We would like to thank you for thoroughly reviewing the manuscript and giving us valuable comments. The number of words in the abstract is less than 300. 

Line no 16-17: In Ethiopia, prevention and treatment of oral health-related illness had been given little attention line no 33: This study revealed a high prevalence of dental carries

Response: The comment is not clear. The statement mentioned in line number 16-17 is for introduction purposes whereas the one mentioned in line number 33 is our finding. 

Line no 36-37: Key words should be arranged alphabetically

Response: According to the comment, Keywords are arranged alphabetically as shown in the track change.

Line no 52 and 53: Even though there is improvement in maintaining oral health worldwide, the problem is still predominant in low-income countries. Also reference for this line.

Response: Based on the comment we have added the reference. 

Line no 204-6 : rephrasing required.

Response: Thank you for the comment; we rephrased it as shown in the track change. 

Recommended read:

1. Zewdu T, Abu D, Agajie M, Sahilu T. Dental caries and associated factors in Ethiopia: systematic review and meta-analysis. Environ Health Prev Med. 2021 Feb 12;26(1):21. doi: 10.1186/s12199-021-00943-3. PMID: 33579186; PMCID: PMC7881546.

2. Teshome A, Andualem G, Derese K. Dental Caries and Associated Factors Among Patients Attending the University of Gondar Comprehensive Hospital Dental Clinic, North West Ethiopia: A Hospital-Based Cross-Sectional Study. Clin Cosmet Investig Dent. 2020;12:191-198

https://doi.org/10.2147/CCIDE.S247179

3. Anguach Shitie, Rahel Addis, Abebe Tilahun, Wassie Negash, "Prevalence of Dental Caries and Its Associated Factors among Primary School Children in Ethiopia", International Journal of Dentistry, vol. 2021, Article ID 6637196, 7 pages, 2021. https://doi.org/10.1155/2021/6637196

Response: Thank you for literatures. We have considered them in the discussion section. 

Reviewer #2: Thank you for the manuscript.

First of all, the anonymity of survey data is not mentioned. Also, what steps did the authors took to safeguard the patient identity.

Response: Thank you for the comment, based on the comment we have included the following in ethical consideration section ‘Written informed consent was also obtained from parents or guardians for children participants. Any information which indicates the Identity of the participants was removed from data collection tool. All information collected from participants was kept confidential’

In 104 line it was mentioned that a structured questionnaire was modified from WHO oral health survey. Could you please elaborate on that? Furthermore, the exact questionnaire should be presented in the manuscript.

Response: For the current study we have adapted a survey tool from WHO (we cited it), We have used the adapted questionnaire to collect socio-demographic characteristics, dietary habits, and factors that affect dental health as shown in the track change. It may not be appropriate for the whole survey tool as it is not ours. 

How many evaluators were there? Were they calibrated?

Response: This comment is not clear. To confirm dental caries we followed the procedure recommended by WHO as indicated in the manuscript.

The uniqueness of the population studied with respect to the other African studies was not discussed in detail. The introduction does not describe in detail the main issue of the paper.

Response: In the discussion, we have mentioned that the ‘study population’ is one reason for the difference in the finding as shown in the track change. 

lines 234-239 - there is no literature reference

Response: The reason we did not include the reference is because it is our comments

line 116 - if the tooth was extracted not at the same clinic and the authors did not have any records of that, how did authors determine the reason of tooth extraction?

Response: We identified by asking past history of participants. 

The authors said that the response rate was 100% and it is hard to believe that all of 288 patients have fully filled the questionnaires.

Response: All participants approached agreed to take part in the study. 

lines 166 - 167 - the authors should change the words into numbers, there has to be consistency throughout the manuscript.

Response: We have corrected accordingly-changed words into numbers

There cannot be a p value equal to 0.000.

Response: It is the value that SPSS gave us during analysis; we have consulted with the statistician and corrected it as <0.001. 

What do the authors consider as a statistically significant result? When COR is <0.05, AOR <0.05 or both?

Response: As indicated in data analysis section, a P-value <0.05 and AOR were considered to determine significant association. 

The discussion section is descriptive, and does not really discuss the results. The results were compared to other studies, however, there were no suggestions how to solve the problems. In the conclusions there are new ideas like integrating promotion of oral health services that has been never mentioned in the discussion, therefore, the conclusions does not really conclude the manuscript. Furthermore, when the results are compared to other studies - where the studies conducted in other countries in the same design? A specification is needed.

Response: Based on the comment we have added the following at the end of the discussion ‘The burden of the dental caries can be reduced by providing proper health education on how to keep oral hygiene and regular visit of dental clinic’. We have also included the nature of study (in the discussion section) we have used for comparison as indicated in the manuscript with track change. 

-We have removed the content claimed from the conclusion section. 

The quality of the language could be better. The authors change past and present times a few times throughout the manuscript.

Response: We have revised and corrected all of them as indicated in the manuscript with track change. 

line 43 - develops instead of develop (corrected)

56 - ‘to’ is missing?

69 - keeping in mind - lowercase letter should be changed in to uppercase at the beginning of the sentence (corrected)

179- index is not a DMF, but DMFT, I would recommend using the newest D3MFT index (corrected)

183- didn’t into did not (corrected)

211- MDFT index (corrected)

Throughout the manuscript different forms of words used - ‘physical examination’ vs ‘intraoral examination’ and they should be unified.

Response: We have corrected all of them.

6. PLOS authors have the option to publish the peer review history of their article (what does this mean?). If published, this will include your full peer review and any attached files.

Do you want your identity to be public for this peer review? For information about this choice, including consent withdrawal, please see our Privacy Policy.

Reviewer #1: No

Reviewer #2: No

---

## [Decision Letter · Decision Letter 1]

16 Feb 2022

PONE-D-21-30095R1Dental caries and associated factors among patients visiting Shashamane Comprehensive Specialized HospitalPLOS ONE

Dear Dr. Ali,

Thank you for submitting your manuscript to PLOS ONE. After careful consideration, we feel that it has merit but does not fully meet PLOS ONE’s publication criteria as it currently stands. Therefore, we invite you to submit a revised version of the manuscript that addresses the points raised during the review process.

We look forward to receiving your revised manuscript.

Kind regards,

Frédéric Denis, Ph.D.

Academic Editor

PLOS ONE

Journal Requirements:

Additional Editor Comments:

Please take into account the comments of reviewer 2 when writing the discussion section.

Reviewers' comments:

Reviewer's Responses to Questions

**Comments to the Author**

1. If the authors have adequately addressed your comments raised in a previous round of review and you feel that this manuscript is now acceptable for publication, you may indicate that here to bypass the “Comments to the Author” section, enter your conflict of interest statement in the “Confidential to Editor” section, and submit your "Accept" recommendation.

Reviewer #1: All comments have been addressed

Reviewer #2: All comments have been addressed

2. Is the manuscript technically sound, and do the data support the conclusions?

Reviewer #1: Yes

Reviewer #2: Partly

3. Has the statistical analysis been performed appropriately and rigorously? 

Reviewer #1: Yes

Reviewer #2: I Don't Know

4. Have the authors made all data underlying the findings in their manuscript fully available?

Reviewer #1: Yes

Reviewer #2: No

5. Is the manuscript presented in an intelligible fashion and written in standard English?

Reviewer #1: Yes

Reviewer #2: No

6. Review Comments to the Author

Reviewer #1: (No Response)

Reviewer #2: Thank you for the correction of the manuscript. However, the Discussion part could be even more improved. Not all of the comments have been addressed properly.

7. PLOS authors have the option to publish the peer review history of their article (what does this mean?). If published, this will include your full peer review and any attached files.

Reviewer #1: No

Reviewer #2: No

---

## [Author Response · Author response to Decision Letter 1]

21 Feb 2022

Additional Editor Comments:

Please take into account the comments of reviewer 2 when writing the discussion section.

Response: Thank you for the comment, we have included study design for studies we have used for comparison we have omitted “Khat is commonly used in Ethiopia for social and pleasure purposes. Also, certain occupational groups like long-distance truck drivers and students during examination time have a high tendency to consume Khat.” From the discussion section. In the previous comment we have been asked to provide reference; however, since it is our idea we are unable to provide reference as a result we omitted it. 

Reviewers' comments:

Reviewer's Responses to Questions

Comments to the Author

1. If the authors have adequately addressed your comments raised in a previous round of review and you feel that this manuscript is now acceptable for publication, you may indicate that here to bypass the “Comments to the Author” section, enter your conflict of interest statement in the “Confidential to Editor” section, and submit your "Accept" recommendation.

Reviewer #1: All comments have been addressed

Reviewer #2: All comments have been addressed

2. Is the manuscript technically sound, and do the data support the conclusions?

Reviewer #1: Yes

Reviewer #2: Partly

3. Has the statistical analysis been performed appropriately and rigorously?

Reviewer #1: Yes

Reviewer #2: I Don't Know

4. Have the authors made all data underlying the findings in their manuscript fully available?

Reviewer #1: Yes

Reviewer #2: No

5. Is the manuscript presented in an intelligible fashion and written in standard English?

Reviewer #1: Yes

Reviewer #2: No

6. Review Comments to the Author

Reviewer #1: (No Response)

Reviewer #2: Thank you for the correction of the manuscript. However, the Discussion part could be even more improved. Not all of the comments have been addressed properly.

Response: Response: Thank you for the comments; we have included study design for studies we have used for comparison purpose. We have omitted “Khat is commonly used in Ethiopia for social and pleasure purposes. Also, certain occupational groups like long-distance truck drivers and students during examination time have a high tendency to consume Khat.” From the discussion section. In the previous comment we have been asked to provide reference; however, since it is our idea we are unable to provide reference as a result we omitted it. (shown in the manuscript with track change.

---

## [Editor Report · Decision Letter 2]

22 Feb 2022

Dental caries and associated factors among patients visiting Shashamane Comprehensive Specialized Hospital

PONE-D-21-30095R2

Dear Dr. Ali,

We’re pleased to inform you that your manuscript has been judged scientifically suitable for publication and will be formally accepted for publication once it meets all outstanding technical requirements.

Kind regards,

Frédéric Denis, Ph.D.

Academic Editor

PLOS ONE
---

## [Editor Report · Acceptance letter]

24 Feb 2022

PONE-D-21-30095R2 

Dental caries and associated factors among patients visiting Shashamane Comprehensive Specialized Hospital 

Dear Dr. Ali:

I'm pleased to inform you that your manuscript has been deemed suitable for publication in PLOS ONE. Congratulations! Your manuscript is now with our production department. 

Kind regards, 

on behalf of

Dr. Frédéric Denis 

Academic Editor

PLOS ONE